# Effect of Beetroot Juice Supplementation on Aerobic Capacity in Female Athletes: A Randomized Controlled Study [note 1]

**DOI:** 10.3390/nu17010063

**Published:** 2024-12-27

**Authors:** Jekaterina Neteca, Una Veseta, Inga Liepina, Katrina Volgemute, Maija Dzintare, Dmitry Babarykin

**Affiliations:** 1Latvian Academy of Sport Education, Riga Stradins University, 16 Dzirciema Street, LV-1007 Riga, Latvia; 2Department of Health Psychology and Paedagogy, Riga Stradins University, 5 J. Asara Street, LV-1009 Riga, Latvia; 3Latvian Academy of Sport Education, Riga Stradins University, 333 Brivibas Street, LV-1006 Riga, Latvia; inga.liepina@rsu.lv (I.L.); katrina.volgemute@rsu.lv (K.V.); maija.dzintare@rsu.lv (M.D.); 4Laboratories and Research Departments, Institute of Innovative Biomedical Technology, 2 Inčukalna Street, LV-1014 Riga, Latvia; dmitry.b@parks.lv

**Keywords:** beetroot juice, aerobic capacity, cardiopulmonary parameter, female athletes

## Abstract

**Background/Objectives:** This study addresses the growing interest in nutritional supplements that improve athletic performance in endurance sports. Previous research suggests that nitrates in beetroot juice enhance blood vessel dilation and oxygen delivery to muscles. However, the effects of these nitrates on cardiopulmonary performance in female athletes remain underexplored. The aim of this study was to evaluate the effect of beetroot juice supplementation on aerobic work capacity in female endurance athletes. **Methods:** A cardiopulmonary exercise test (CPET) was conducted to assess aerobic work capacity. Eighteen healthy female endurance athletes (22.9 ± 5.6 years) participated in the study. The participants were randomly assigned to two groups: the control group (placebo group *n* = 9), which received a nitrate-free placebo beverage, and the experimental group (beetroot juice group *n* = 9), which consumed 50 mL of beetroot juice concentrate (~6.2 mmol nitrate) two and a half hours before the second test. **Results:** The results showed that the beetroot juice group demonstrated significant improvements in minute ventilation (VE), respiratory equivalents (VE/VO_2_ and VE/VCO_2_), and heart rate (HR) (*p* < 0.05). Maximal oxygen consumption (VO_2_ max) increased by 4.82% in the beetroot juice group (from 35.24 ± 5.07 to 36.94 ± 4.91 mL·min^−1^·kg^−1^), whereas a small decrease was observed in the placebo group. **Conclusions:** These findings indicate that beetroot juice may be an effective ergogenic aid, enhancing oxygen utilization and energy production during exercise in female athletes. In terms of practical applications, beetroot juice could contribute to improved athletic performance and serve as a valuable addition to athletes’ nutritional plans. Future studies should explore the long-term effects, optimal dosages, and duration of supplementation in larger and more diverse populations.

## 1. Introduction

Nutritional supplements have become an important part of sports nutrition because they can significantly affect athletic performance, recovery, and overall health. Research in recent years shows that athletes are increasingly using various nutritional supplements to improve performance, increase endurance and promote faster recovery [1,2]. According to recent analyses, there is increasing research into how these dietary supplements, such as nitrates found in beetroot juice, could improve athletic performance by improving oxygen availability and blood flow to muscles [3,4].

In endurance sports, the efficient use of oxygen and the proper functioning of the cardiovascular system are particularly important, as athletes need high aerobic endurance for the best results. The use of beetroot juice, due to its high nitrate content, can improve the production of nitric oxide, which in turn dilates blood vessels and improves oxygen supply to muscles [5,6,7].

Studies show that regular consumption of beetroot juice can reduce systolic blood pressure, improve blood flow and increase maximal oxygen consumption (VO_2_ max) [8]. These characteristics are especially important in endurance sports, where every percentage point of improvement can affect an athlete’s performance in competition. Recent evidence suggests that NO_3_- supplementation may be more beneficial for augmenting high-intensity and intermittent exercise which induces local hypoxia within the muscle, since the NO_3_–NO_2_–NO pathway is stimulated under conditions of low pH and low oxygen availability [9,10]. A recent study [11] confirms that consumption of beetroot juice effectively improves exercise capacity and cardiovascular function in healthy men. However, to better understand the combined effects of beetroot juice consumption and exercise, additional studies on the effects of exercise and the cardiovascular system are needed, especially taking into account gender, age, aerobic endurance and environmental factors [11].

Unfortunately, research on female athletes is limited: women make up only 4–13% of all participants in sports science studies [2]. This underrepresentation of women in research means that nutritional and training recommendations are often based on data from men, which ignores the physiological differences and needs essential to women’s athletic performance. Only a small number of studies have looked at the response of female athletes to the use of beetroot juice. Some studies suggest that women may benefit more from nitrates because they have a higher proportion of oxidative muscle tissue and a more efficient capacity to metabolize nitrates [12].

Gender differences in physiology that affect aerobic work capacity [13,14,15,16] determine not only differences in performance but also responses to training and supplements. In a study by [17], it was found that aerobic work capacity is lower during the luteal phase of the menstrual cycle compared to the follicular phase. Other researchers have noted increased minute ventilation (VE) during the luteal phase while respiratory muscle oxygen consumption (VO_2_) increases and contributes to an increase in total VO_2_, which may be explained by additional work demands when progesterone levels are elevated. Although there are strong correlations between progesterone levels and increases in VE, the exact mechanisms have not yet been identified [17,18,19].

To investigate female athletes’ performance and responses to nutritional supplementation, studies should consider hormonal fluctuations during the menstrual cycle that may affect physical performance and responses to training or supplementation [16]. Hormones such as estrogen and progesterone can affect muscle function, strength, and endurance, thus altering physical performance during different phases of the cycle [20]. In addition, women may metabolize carbohydrates and fats differently, which affects their ability to use energy during prolonged exercise [21].

Research suggests that women may have a higher aerobic endurance potential, but its expression is related to hormonal factors and phases of the menstrual cycle [22]. Accurate results in research on women also require specific methodology for endurance and recovery [23].

The effects of dietary supplements such as nitrates in women may vary depending on hormonal status, so it is important to analyze female responses compared to male responses [4]. The analysis of cardiopulmonary indicators is important for understanding the physical fitness of athletes. Studies show that beetroot juice-stimulated nitric oxide production, that can improve cardiac efficiency and blood flow, which are essential for aerobic work capacity [5]. An analysis of cardiopulmonary indicators helps to accurately understand the effect of exercise and nutritional supplements in sports performance [24].

Given the potential benefits of beetroot juice, it is important to investigate its effects on aerobic performance in female athletes. Research suggests that women may benefit more from nitrates because their muscle tissue is dominated by more oxidative muscle types than men [7], so studies evaluating the effects of nitrates in women with different fitness levels and gender-specific physiology are needed [12].

This study followed the recommendation to account for menstrual cycle phases and exclude women using hormonal contraceptives when investigating the effects of dietary nitrates (NO_3_^−^), as suggested by [25]. While this approach limited the study’s scope, it was crucial because hormonal fluctuations during the menstrual cycle can influence nitric oxide production and how the body responds to dietary nitrates. Estrogen, in particular, can significantly affect the body’s ability to convert NO_3_^−^ into its active form, nitric oxide, which plays an important role in enhancing vascular function.

The aim of this study was to evaluate the effect of beetroot juice supplementation on aerobic work capacity in female endurance athletes.

**Hypothesis:** 
*Consuming beetroot juice concentrate with approximately 6.2 mmol of nitrate two and a half hours prior to a cardiopulmonary exercise test (CPET) will lead to more efficient oxygen utilization and improvements in aerobic work capacity indicators, such as VO_2_ max, minute ventilation (VE), heart rate (HR), and respiratory equivalents (VE/VO_2_, VE/VCO_2_), in female endurance athletes compared to the placebo group.*


## 2. Materials and Methods

### 2.1. Study Design

The present study utilized a randomized controlled trial (RCT) design to assess the effects of beetroot juice supplementation on aerobic performance in female athletes. The RCT design was selected due to its established robustness and reliability in clinical research, as it enables the control of confounding variables and ensures the validity and reproducibility of results. This approach is particularly advantageous in assessing interventions where randomization minimizes selection bias and increases internal validity. Participants were randomly assigned to one of two groups: the experimental group (beetroot juice supplementation group, BJG) or the control group (placebo group, PLG). The BJG received a nitrate-rich beetroot juice intervention, while the PLG received a placebo beverage containing no nitrates. This design allows for a comparison between the effects of beetroot juice and a neutral placebo, providing insight into its potential benefits for endurance performance. Both groups followed identical protocols, ensuring that the only difference between them was the supplementation provided.

### 2.2. Participants

Inclusion Criteria: Healthy females aged 18–42 years, not pregnant or breastfeeding, with no medical conditions affecting exercise performance or cardiorespiratory data interpretation, and with regular involvement in amateur endurance sports (running, swimming, or cycling) for at least three months prior to the study; engaging in moderate-intensity aerobic exercise (30–60 min per session, 60–75% HRmax) performed 3–5 times per week; no cardiovascular or respiratory diseases, as confirmed by a health screening questionnaire and initial assessment; refrained from medications or supplements affecting cardiovascular or metabolic responses (e.g., beta-blockers or nitrates).

Exclusion Criteria: Participants were excluded from the study if they had experienced any acute illness within the past month or were taking medications that might affect cardiorespiratory function, as such conditions could interfere with the measurement accuracy. Pregnant women were also excluded due to potential health risks associated with the physical exertion required for the tests. Furthermore, to control for hormonal influences, participants using hormonal contraception were excluded from the study.

Of the 26 participants initially recruited, 5 did not meet the inclusion and exclusion criteria. Consequently, 21 participants were deemed eligible and provided informed consent to participate. Fitness levels, as measured by VO_2_ max, ranged from 35.24 ± 5.07 to 36.94 ± 4.91 mL·min^−1^·kg^−1^. Anthropometric characteristics were as follows: mean age 22.9 ± 5.6 years, height 165.8 ± 7.14 cm, weight 64.0 ± 0.57 kg. The participants were mainly university students with a sedentary lifestyle outside structured sports. They did not engage in other recreational or physically demanding activities, ensuring consistent physical activity levels. Their daily schedules were focused on academic responsibilities. They practiced endurance sports during their free time, with weekly training volumes ranging from 150 to 300 min in the moderate-intensity heart rate zone. Warm-up and cool-down periods made up about 30% of the total session time but were not included in the target zone. The participants had adopted this training regimen 3 months to 1 year before the study, ensuring they were still in the adaptation phase of endurance training, minimizing the impact of long-term, high-intensity training on results.

Data collection was conducted from 27 October 2023 to 14 April 2024. The duration was extended to accommodate the individualized participation schedule for each participant, which was planned based on the follicular phase of their menstrual cycle. This approach was adopted to enhance the reproducibility and interpretability of studies on dietary nitrate (NO_3_^−^), as research suggests menstrual cycle phases may influence exercise performance and metabolic responses [25].

The study was conducted in accordance with the Declaration of Helsinki and received approval from the relevant Ethics Committee (protocol code Nr.2/51813, 28 October 2021). Prior to the commencement of the study, all participants were fully informed about the study’s objectives, the beetroot juice intervention, the study procedures, and the guidelines to follow concerning participation, exercise testing, contraindications, and preparation requirements. The key guidelines included:The last meal should occur no later than 3 h prior to the test;Smoking, the use of medication or nutritional supplements, mouthwash, and chewing gum were prohibited;High-intensity physical activities were prohibited 24 h before the test;Participants were required to wear comfortable sports clothing and footwear.

Smoking was prohibited before and during the exercise test due to its effects on airway constriction and increased carbon monoxide levels in the blood, which could negatively affect exercise performance by reducing oxygen delivery to the muscles [26].

Participants who agreed to take part in the study signed an informed consent form confirming that they met the inclusion criteria, had been briefed about the study protocol, agreed to follow the instructions, and understood that they could withdraw from the study at any time without any consequences.

#### Pre-Test Measurements

To tailor an individual ergometer test protocol for each participant, which is essential for determining aerobic capacity and oxygen consumption, the following measurements were taken prior to each test:Height measurement: The participant stood upright with feet together, pressing the heels, back, chest, and back of the head against the wall. Height was recorded in centimeters using a tape measure.Body weight measurement: The participant stood on a scale in a T-shirt and shorts. Body weight was recorded in kilograms.

### 2.3. Instruments

Aerobic work capacity was determined via a cardiopulmonary exercise test (CPET) conducted on an exercise bike with electronic brakes (Lode Excalibur Sport, Groningen, The Netherlands), using the “Vyntus CPX” system for breath gas analysis.

The CPET parameters included: VO_2_ (oxygen consumption), VE (ventilation or minute ventilation), HR (heart rate), VE/VO_2_ (ratio of ventilation to oxygen consumption), VE/VCO_2_ (ratio of ventilation to carbon dioxide expiration), O_2_ pulse (oxygen pulse), which reflects the heart and lung response to physical exertion [27].

During CPET, inspiratory and expiratory pulmonary gas exchange data were continuously collected and averaged over 10-s intervals (Vyntus CPX metabolic cart, Vyaire Medical, Chicago, IL, USA). A mask connected to a spirometer (Vyaire Medical GmbH, Höchberg, Germany), which measures breathing parameters, including oxygen and carbon dioxide concentrations, was worn by the participants. Respiratory parameters were recorded, and inspiratory/expiratory gas analysis was performed, with pulse oximetry also recorded. The peak VO_2_ was defined as the highest mean VO_2_ achieved during any 30-s period before the CPET concluded.

### 2.4. Test Protocol

The RCT protocol included two CPET tests, each performed using the Vyntus CPX cardiopulmonary measurement device (Vyaire Medical GmbH, Höchberg, Germany). Typically, CPETs last 8 to 12 min, depending on the participant’s exercise intensity. Given the endurance sports background of the participants, the duration was extended to 15 min [28]. The starting power of the cycle ergometer (in watts) was calculated based on anthropometric measurements.

Test Procedure:Warm-up: 15–20 min of individualized warm-up, without the cycle ergometer.CPET: 15 min on the cycle ergometer.

The CPET protocol began with 5 min of rest measurements, followed by a 3-min warm-up on the cycle ergometer. The intensity was then progressively increased by 0.2 W/kg/min every 3 min. At the start of the CPET, participants were instructed to maintain a steady pedaling speed of 70–75 rpm at a constant workload.

The second CPET was performed one week after the first test. Prior to the second CPET, participants were instructed to refrain from chewing gum, using menthol mouthwash, taking medications, or consuming dietary supplements, to avoid potential interference with nitrate (NO_3_^−^) activity. Menthol mouthwashes were specifically excluded, as they have been shown to impair muscle oxygenation post-exercise, potentially by reducing nitric oxide production and affecting vasodilation [29]. The test procedure is shown in Figure 1.

### 2.5. Intervention

Two and a half hours prior to the second CPET, participants in the BJG consumed 50 mL of nitrate-rich beetroot juice concentrate (NO_3_^−^), while participants in the PLG consumed an equivalent volume of nitrate-free beetroot juice concentrate. The concentrate, derived from red beet juice and processed using validated membrane fractionation technology (Institute of Innovative Biomedical Technology Ltd., Riga, Latvia), contained 3421 ± 445 mg/kg of nitrates, 1% fructose, 1% glucose, and 3.5 ± 0.1% sucrose. Each 50 mL serving provided approximately 6.2 mmol of nitrates (NO_3_^−^). This dosage was selected based on prior research [30], which demonstrated that dietary nitrate supplementation enhances physical performance by improving oxygen efficiency and reducing exercise-induced fatigue.

Following the same warm-up protocol, participants performed the second CPET under identical conditions as the first test.

### 2.6. Data Exclusion

Participants who did not complete the CPET or who encountered technical issues, such as equipment failure or participant withdrawal, were excluded from the final analysis. A total of three participants were excluded from the study based on these criteria. Specifically, one participant was excluded for not completing the test, another for non-adherence to the study protocol, and a third due to technical difficulties. The study design flowchart, which illustrates the progression of participants throughout the study, is shown in Figure 2.

#### Data Analysis

The Statistical Package for the Social Sciences (SPSS), version 29.0 and Microsoft Office Excel for Microsoft 365 MSO (Version 2402 Build 16.0.17328.20670) 64-bit were used for data analysis. In the first step, the collected data were carefully reviewed and summarized for further statistical analysis. Statistical methods were selected based on the research objectives. Descriptive statistics were initially applied to analyze the obtained data from test results. To assess changes in variables over time between the first and second testing phases within a randomized controlled trial, a one-way repeated-measures ANOVA was performed. The analysis incorporated two groups and two testing points with repeated measurements. Prior to conducting the ANOVA, the normality of the data distribution was evaluated using the Kolmogorov-Smirnov test. Additionally, a test of sphericity was conducted to ensure accurate interpretation of the results. Effect sizes for repeated (paired) measurements were calculated using Cohen’s *d* values [31]. A minimally significant difference in performance was defined as less than 0.6%, based on the guidelines by Hamilton (2006) [32]. Statistical significance was established at *p* < 0.05.

## 3. Results

This study was a RCT of virtually healthy women participating in endurance sports. Participants were randomly assigned to two groups: placebo group (PLG, *n* = 9) and beetroot juice group (BJG, *n* = 9). Both groups, PLG and BJG, were nearly identical in their baseline characteristics. The initial data, including age, height, weight, and other relevant parameters, showed no statistically significant differences between the groups, ensuring that any observed outcomes could be attributed to the intervention rather than pre-existing disparities. The first cardiopulmonary exercise test (CPET) was performed in both groups according to a uniform protocol. A week after the first test, a second test was performed, before which BJG consumed a beetroot juice concentrate containing ~6.2 mmol NO_3_^−^ and PLG consumed a beetroot juice concentrate containing no nitrates.

To evaluate the effect of beetroot juice supplementation on aerobic work capacity, the following parameters were analyzed during CPET: maximal oxygen consumption (VO_2_ max), heart rate (HR), minute ventilation (VE), ventilation to oxygen consumption ratio (VE/VO_2_) and ventilation and expiratory carbon dioxide ratio (VE/VCO_2_). The cardiopulmonary parameters analyzed during CPET tests are summarized in Table 1.

The results showed that BJG VO_2_ max increased by 4.82% (from 35.24 ± 5.07 to 36.94 ± 4.91 mL·min^−1^·kg^−1^), while it decreased by 0.57% in the PLG group (from 35.06 ± 4.87 to 34.86 ± 5.01 mL·min^−1^·kg^−1^) between the first and second tests.

Statistically significant changes in mean heart rate in BJG and PLG after test 1 and test 2 were observed and are shown in Figure 3. Mean heart rate (HR) in the BJG group changed significantly after the second test (*p* < 0.05), decreasing from 165 ± 9 to 162 ± 10 bpm. On the other hand, in the PLG group, HR increased from 160 ± 9 to 164 ± 8 beats per minute after the first and second tests. Vertical bars represent 95% confidence intervals of the mean.

In the BJG group, VE (L/min) values changed significantly after the second test (*p* = 0.006), decreasing from 76.56 ± 19.39 to 70.22 ± 17.48 (d = 1.1). On the other hand, in the PLG group, VE scores after the second test increased from 70.78 ± 12.25 to 76.44 ± 14.59 (d = 1.6, *p* = 0.001). Figure 4. Statistically significant (*p* < 0.05) changes in pulmonary ventilation (VE, L/min) in BJG and PLG groups after the first and second test.

The ventilatory equivalents for oxygen (VE/VO_2_) increased by 0.84% in the PLG group (from 2.21 ± 0.38 to 2.23 ± 0.38; d = 1.6, *p* = 0.001), while in the BJG group, VE/VO_2_ decreased from 2.10 ± 0.49 to 1.90 ± 0.38 (d = 1.1, *p* = 0.006) (see Figure 5).

The respiratory equivalent: ventilation to expiratory carbon dioxide ratio (VE/VCO_2_) is graphically represented for the BJG and PLG groups (Figure 6). In the BJG group, mean VE/VCO_2_ values decreased significantly (*p* = 0.025) from 0.030 ± 0.00 to 0.028 ± 0.00 (d = 0.9) after the second test. The effect size of this change is important, and it is large (0.8).

The results of the study show that the use of beetroot juice can contribute to the improvement of aerobic work capacity in female athletes by improving oxygen consumption, more effectively regulating ventilation and heart rate during exercise compared to the placebo group.

## 4. Discussion

This study was a RCT involving 18 virtually healthy women participating in endurance sports. The participants were divided into two groups: one group consumed 50 mL of beetroot juice concentrate containing ~6.2 mmol nitrate, and the other a placebo beetroot juice without nitrate. The aim of this study was to evaluate the effect of beetroot juice supplementation on aerobic work capacity in female endurance athletes. Hypothesis: Consuming beetroot juice concentrate with approximately 6.2 mmol of nitrate two and a half hours prior to a cardiopulmonary exercise test (CPET) will lead to more efficient oxygen utilization and improvements in aerobic work capacity indicators, such as VO_2_ max, minute ventilation (VE), heart rate (HR), and respiratory equivalents (VE/VO_2_, VE/VCO_2_), in female endurance athletes compared to the placebo group.

The results of the study supported the hypothesis that beetroot juice intake provides positive changes in several physiological parameters. There was a trend towards significant changes in maximal oxygen consumption (VO_2_ max), which increased by 4.82% in the BJG group (from 35.24 ± 5.07 to 36.94 ± 4.91 mL-min^−1^-kg^−1^), indicating improved oxygen availability and blood flow to muscle [3,4] and potentially better ability to withstand prolonged intense exercise. These findings align with those of [33], who reported improved VO_2_ max and oxygen economy in response to dietary nitrates, although their cohort included both men and women. Moreover, ref. [34] also documented similar VO_2_ max improvements in participants with moderate aerobic capacity, further supporting the current study’s outcomes.

VO_2_ max is in the range of 30–85 mL-min^−1^-kg^−1^ in healthy adults, covering the spectrum of aerobic performance from untrained to elite endurance athletes [35]. VO_2_ max is an important marker of aerobic work capacity and its increase is associated with higher fitness and reduced risk of cardiovascular disease [36,37]. This change is particularly important given that even small increases in VO_2_ max (e.g., +3.5 mL/min/kg) can reduce cardiovascular risk by more than 13% [38]. Contrastingly, ref. [30] noted less pronounced VO_2_ max improvements among elite athletes, possibly due to physiological saturation. This highlights that individuals with moderate fitness levels, as in the BJG group, may derive greater relative benefits. These results are in line with previous research indicating that nitrates found in beetroot can improve muscle oxygenation and use it more efficiently [4,5]. In contrast, VO_2_ max decreased by 0.57% (from 35.06 ± 4.87 to 34.86 ± 5.01 mL·min^−1^·kg^−1^) in the placebo group, indicating that nitrate supplementation is a significant factor in improving oxygen consumption.

In addition to VO_2_ max, significant changes in other parameters were also observed. The decrease in minute ventilation (VE) in the BJG group was statistically significant (*p* = 0.006), indicating improved lung efficiency by reducing ventilation at the same exercise intensity. This indicates that women who consumed beetroot juice performed physical tests with less respiratory effort, which helped prevent fatigue and improve endurance during prolonged exercise. This reduction in VE mirrors findings from [39], who emphasized the role of dietary nitrates in optimizing oxygen utilization and minimizing respiratory strain during high-intensity activity. VO_2_ max is higher in BJG than in PLG because NO, which is formed from nitrites and nitrates in beetroot juice, relaxes vascular smooth muscle and dilates blood vessels more. As a result, more blood flows to the muscles, and they receive more O2, glucose, etc. Improved muscle oxygen delivery provides ergogenic benefits [37].

Another important indicator was the ventilatory equivalent VE/VO_2_. The ratio of ventilation to oxygen consumption decreased by 9.52% in the BJG group, indicating a more efficient use of oxygen. This result is consistent with previous research showing that nitrates help optimize muscle oxygen utilization, especially during high-intensity exercise [30]. On the contrast, in the placebo group, the VE/VO_2_ value increased by 0.84%, indicating a lower ventilation efficiency.

In addition, significant improvement was also observed in ventilatory equivalent VE/VCO_2_ scores and was statistically significant in the BJG group (*p* < 0.025). VE/VCO_2_ decreased after the second test in the BJG group from 0.030 ± 0.00 to 0.028 ± 0.00 (d = 0.9, *p* = 0.025), indicating more efficient gas exchange and CO_2_ elimination. On the contrary, this indicator increased in the PLG group, indicating a lower efficiency of the respiratory system at a similar load (Figure 6). This improvement is essential under conditions of prolonged exercise, where higher ventilation efficiency helps to reduce fatigue and maintain performance [37,38].

In addition, the decrease in HR in the BJG group was statistically significant (*p* = 0.001), indicating improved cardiac efficiency. This suggests that the heart was able to do more work at lower exercise intensities, which is essential in long-term endurance sports. These results are consistent with previous studies in which nitrates have been shown to be effective in improving blood flow and reducing oxygen demand during high-intensity exercise [33,40,41]. Similar results are found in a study by [42], who reported a significant decrease in HR in a high-intensity intermittent running test after 6 days of beetroot juice consumption. In this study, the mean HR was lower in the BJG group (172 ± 2) compared to the placebo group (175 ± 2; *p* = 0.014). The authors concluded that six days of beetroot juice intake effectively improved high-intensity interval training performance in trained soccer players. These research results indicate the possible potential of beetroot juice to contribute to the improvement of cardiovascular function also in high-intensity exercise [42]. Scientists observed similar results in research [43], when acute nitrate ingestion led to significant decreases in the mean HR during high-intensity interval exercise.

This study followed the recommendation to consider menstrual cycle phases and exclude hormonal contraceptive users in dietary NO_3_^−^ studies of women [25]. This strategy limited the study, but it was important because it should be taken into account that the hormonal changes that occur in women both during the menstrual cycle can affect the synthesis of nitric oxide and affect how the body responds to dietary nitrates. Hormonal changes associated with estrogen levels can significantly alter the body’s ability to reduce NO_3_^−^ to its biologically active form, which helps improve vascular function [25].

Limitations of this study include the small sample size, which could limit the generalizability of the findings. In addition, in order to more accurately assess the effect of beetroot juice on the physiology of the body, it would have been useful to quantify the concentration of nitrates and nitrites in the plasma. This would help to better understand the metabolism of these substances and its effect on physiological parameters. Future studies would need to include larger sample sizes of participants as well as use more detailed analysis tools to more accurately assess the potential of beetroot juice as an ergogenic supplement.

This study demonstrates that beetroot juice consumption is an effective strategy for endurance athletes to improve aerobic performance such as VO_2_ max, ventilatory efficiency and cardiac output. The nitrates found in beetroot improve the use of oxygen in the muscles, which in turn contributes to the improvement of prolonged exercise. Although the study has limitations, it provides important evidence that can serve as a basis for future research, especially regarding gender differences and hormonal factors that may influence nitrate exposure in women.

## 5. Conclusions

This study investigated the effects of beetroot juice supplementation on aerobic work capacity and cardiopulmonary performance in female endurance athletes. The results demonstrated that the consumption of beetroot juice led to a 4.82% increase in VO_2_ max, a significant improvement in ventilation efficiency (VE/VO_2_ and VE/VCO_2_), and a reduction in heart rate, reflecting enhanced cardiovascular and respiratory function. These findings indicate that beetroot juice consumption can reduce fatigue during prolonged exercise by improving oxygen utilization and energy efficiency, which are critical for endurance sports performance. The results revealed distinct differences between the beetroot juice group (BJG) and the placebo group (PLG), emphasizing the potential of nitrates to optimize physiological responses during exercise in female athletes. The BJG exhibited significant improvements in ventilation and gas exchange efficiency, while the PLG showed signs of decreased ventilatory efficiency and a slight decline in VO_2_ max. These differences underscore the role of targeted supplementation in enhancing performance, particularly for female athletes who may benefit from gender-specific strategies due to physiological and hormonal differences.

Although the results of this study are promising, they also highlight the need for further research focusing on long-term supplementation effects, optimal dosing strategies, and the influence of hormonal phases during the menstrual cycle. This is critical to ensure that women are adequately represented in sports nutrition research and that their unique physiological and hormonal needs are addressed.

### Practical Implications of This Study

The results of this study have practical implications for coaches, athletes, and sports professionals. The methodology provides a framework for incorporating beetroot juice into an athlete’s daily nutritional routine as a natural strategy to enhance performance during intense physical activity. These findings are particularly relevant for endurance sports, where evidence supports the inclusion of beetroot juice as a beneficial dietary supplement. Importantly, this study highlights significant benefits for female athletes, demonstrating that beetroot juice can effectively enhance both performance and ergogenic effects, making it a valuable addition to the nutrition plan of athletes seeking to optimize their results.

## Figures and Tables

**Figure 1 nutrients-17-00063-f001:**
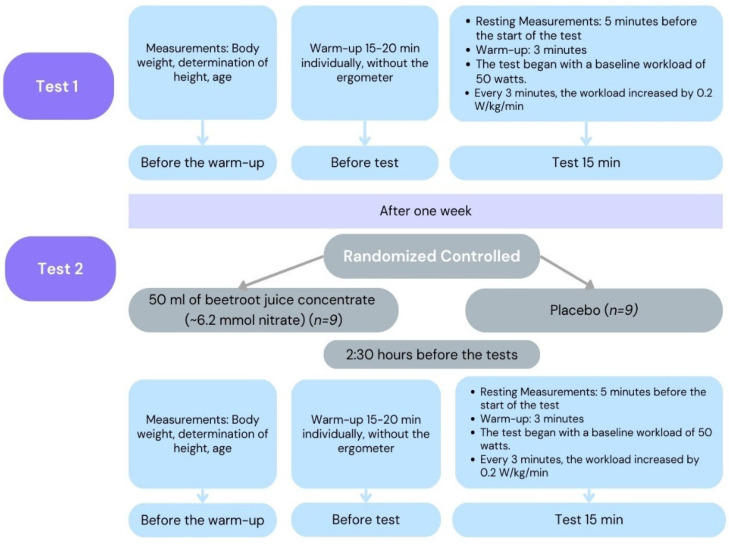
Flow chart of the test procedure.

**Figure 2 nutrients-17-00063-f002:**
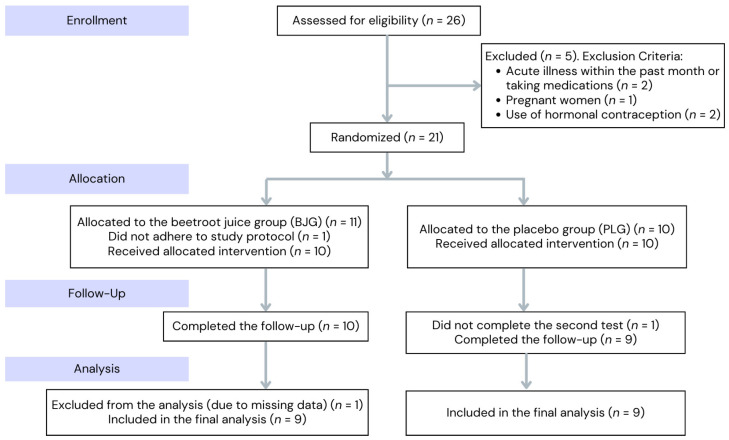
Flow chart of the study design.

**Figure 3 nutrients-17-00063-f003:**
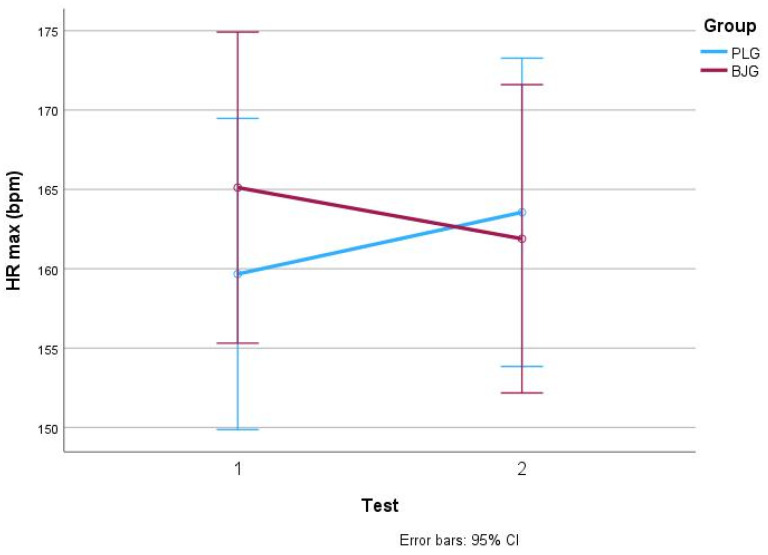
HR max values in BJG and PLG (Test 1—without beet root juice, test 2—after consuming 50 mL beetroot juice concentrate or placebo).

**Figure 4 nutrients-17-00063-f004:**
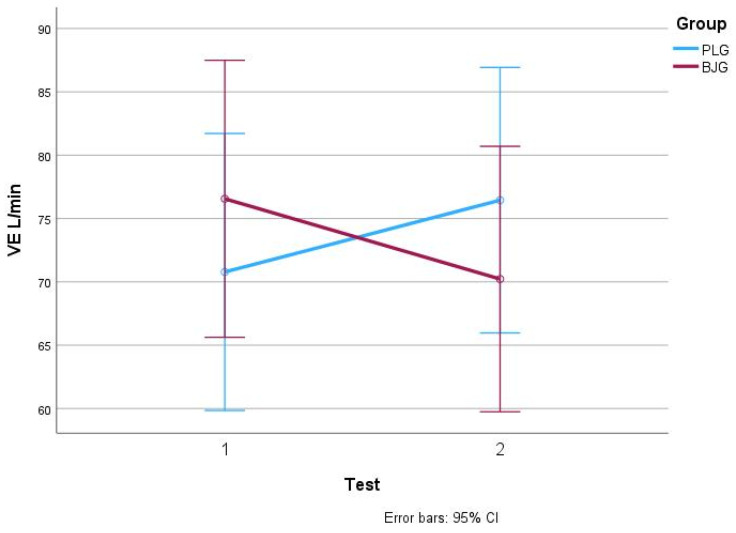
VE L/min values in BJG and PLG (Test 1—without beet root juice, test 2—after consuming 50 mL beetroot juice concentrate or placebo).

**Figure 5 nutrients-17-00063-f005:**
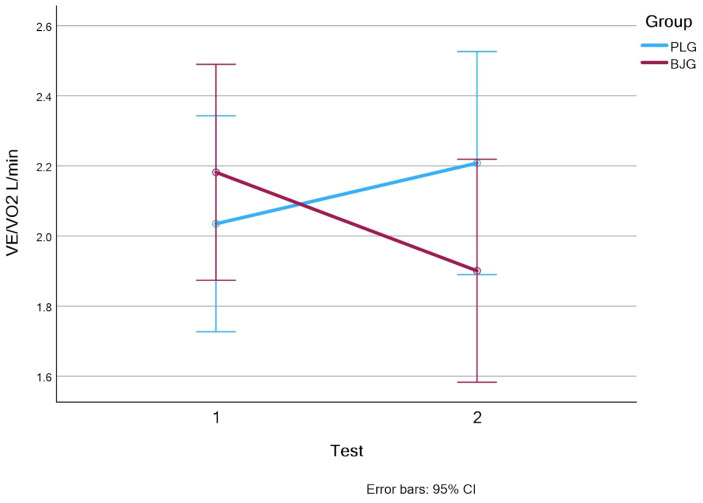
The ventilatory equivalents for oxygen VE/VO_2_ L/min values in BJG and PLG (Test 1—without beet root juice, test 2—after consuming 50 mL beetroot juice concentrate or placebo).

**Figure 6 nutrients-17-00063-f006:**
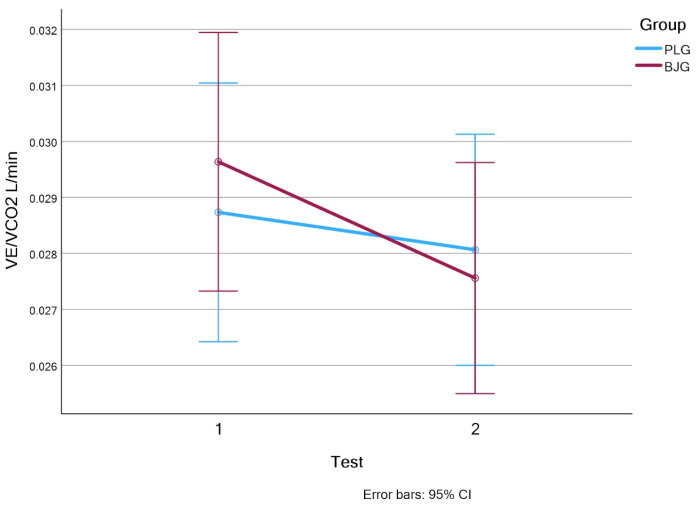
The ventilatory equivalents carbon dioxide VE/VCO_2_ L/min values in BJG and PLG (Test 1—without beet root juice, test 2—after consuming 50 mL beetroot juice concentrate or placebo).

**Table 1 nutrients-17-00063-t001:** Cardiopulmonary parameters and effect sizes in PLG and BJG across two tests.

Group	Measurment	MV	SD	(*n*)	*p* Value	Effect Size (T1–T2, Cohen’s Correlation)	95% Confidence Interval for the Effect	
							Lower Limit	Upper Limit
PLG	VE Test 1	70.78	12.25	9	0.001	−1.598	−2.549	−0.612
	VE Test 2	76.44	14.59	9				
	HR max Test 1	160	9	9	0.084	−0.595	−1.236	0.076
	HR max Test 2	164	8	9				
	Test 1 VO_2_max	35.06	4.87	9	0.214	0.406	−0.226	1.016
	Test 2 VO_2_max	34.86	5.01	9				
	VEVCO_2_Test1	0.029	0.00	9	0.406	0.264	−0.347	0.860
	VEVCO_2_Test2	0.028	0.00	9				
	VEVO_2_Test1	2.03	0.34	9	0.001	−1.601	−2.553	−0.614
	VEVO_2_Test2	2.21	0.38	9				
BJG	VE Test 1	76.56	19.39	9	0.006	1.100	0.289	1.875
	VE Test 2	70.22	17.48	9				
	HR max Test 1	165	9	9	0.001	1.567	0.593	2.507
	HR max Test 2	162	10	9				
	Test 1 VO_2_max	35.24	5.07	9	0.058	−0.666	−1.322	0.021
	Test 2 VO_2_max	36.94	4.91	9				
	VEVCO_2_Test1	0.030	0.00	9	0.025	0.827	0.098	1.522
	VEVCO_2_Test2	0.028	0.00	9				
	VEVO_2_Test1	2.18	0.49	9	0.006	1.100	0.90	1.875

Note: MV = arithmetic mean; SD = standard deviation; *n* = number of participants correlation is significant at the 0.05 level (2-tailed).

## Data Availability

Data have been collected as part of the PhD thesis and will be full version publicly available upon completion of the PhD. Further inquiries can be directed to the corresponding authors.

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
