# Peer review of "Effect of Beetroot Juice Supplementation on Aerobic Capacity in Female Athletes: A Randomized Controlled Study†"

_nutrients, 2024, doi:10.3390/nu17010063_

Round 1
Reviewer 1 Report
Comments and Suggestions for Authors
see attached document

Author Response
For research article
Response to Reviewer X Comments
|
||
1. Summary |
|
|
Thank you very much for taking the time to review this manuscript. Please find the detailed responses below and the corresponding revisions/corrections highlighted in the re-submitted files.
|
||
2. Questions for General Evaluation |
Reviewer’s Evaluation |
Response and Revisions |
Does the introduction provide sufficient background and include all relevant references? |
Yes |
|
Are all the cited references relevant to the research? |
Yes |
|
Is the research design appropriate? |
Yes |
|
Are the methods adequately described? |
Yes |
|
Are the results clearly presented? |
Can be improved |
|
Are the conclusions supported by the results? |
Can be improved |
|
Feedback for article entitled: Effect of Beetroot Juice Supplementation on Aerobic
Capacity in Female Athletes: A Randomized Controlled Study
This article was well written. A clear difference to this RCT examining Beetroot juice
supplementation effect on endurance performance in athletes was the focus of the
cohort being female athletes.
The design was simple and compared one intervention, beetroot juice (nitrate 6.2
mmol) vs placebo (no nitrate in the juice). This design allowed for investigating potential
benefits for endurance performance in female athletes.
Thank you for your positive feedback and recognition of the study's design and focus. We are glad that you found the study well-written and appreciated its emphasis on female athletes, as this demographic is often underrepresented in sports science research. We agree that the straightforward design, comparing beetroot juice (nitrate 6.2 mmol) with a nitrate-free placebo, was a key strength of the study. This simple yet robust approach allowed us to investigate the potential benefits of nitrate supplementation on endurance performance in female athletes while minimizing confounding factors. Thank you again for your encouraging comments and for highlighting these aspects of our work.
- Point-by-point response to Comments and Suggestions for Authors
Some comments/questions
- The introduction was very long and would benefit in being reduced. There was
repetition of information/background in research related to female athletes and
performance and what is lacking and what should be done.
Response: Thank you for your valuable feedback regarding the introduction. We agree that the introduction was overly long and included repetitive information about research related to female athletes and performance. To address this, we have revised the introduction to reduce its length and eliminated redundant information. These changes have been highlighted in the revised manuscript for your review.
- Line 134/135: women using hormonal contraception should be included in the
exclusion criteria
Response: Thank you for your logical suggestion. We have moved the sentence from lines 134/135 to the exclusion criteria.
- Are participants told not to take nitrate rich foods in the last meal (line 142) and
also in regard to food eaten the week between the first and second CPET? This
would cover variables such as diet also having impact on parameters being
measured?
Response: Thank you for pointing this out. Before the start of the study, all participants were fully informed about the study objectives, the beetroot juice intervention, the study procedures, and the guidelines to be followed. Participants were asked to follow their usual diet, without any specific instructions to modify their intake, including regarding nitrate-rich foods. This means that participants were not required to make any particular dietary restrictions concerning nitrate-rich foods, and they were asked to maintain their regular diet in order to avoid any artificial impact on the study results.
- For the test protocol: at what stage of menstrual cycle was the test/intervention
done? This is not mentioned in your study and is one of the big factors
highlighted in the introduction.
Response: Thank you for pointing this out. We agree that the phase of the menstrual cycle is an important factor that can influence the results of the study. This information is included in our manuscript, where we mention that the testing schedule was planned based on the follicular phase of each participant’s individual menstrual cycle. This approach was adopted to enhance the reproducibility and interpretability of the study, as research suggests that menstrual cycle phases can influence exercise performance and metabolic responses (Baranauskas, Freemas, Tan, & Carter, 2022).
- Change Figure 2 to be correct CONSORT diagram for RCTs.
Response: Thank you for pointing this out. We agree with this comment and have updated Figure 2 to reflect the correct CONSORT diagram for RCTs.
- Reference for the effect size (cohen d values) needs to be included – line 221
Response: Thank you for the recommendation. We agree and have added the reference: "Effect sizes for repeated (paired) measurements were calculated using Cohen's ? values (Cohen, 2013)."
- Reference 33 cited – Line 222; it is unclear what this reference provides support
for when you say “based on the guidelines” in the reference cited? Please
explain.
Response: Thank you for bringing this to our attention. We acknowledge the oversight and would like to clarify that the citation was incorrectly used due to a formatting error. The intended statement should read: "A minimally significant difference in performance was defined as less than 0.6%, based on the guidelines by Hamilton (2006) (Hamilton, 2006)." Hamilton's guidelines specifically define this threshold as a criterion for determining minimally significant performance differences. The corrected statement now ensures that the citation appropriately supports this claim.
- Line 224 and 277 – you say “virtually healthy woman” --- what do you mean?
Response: Thank you for pointing this out. By "virtually healthy women," we refer to individuals who do not have any diagnosed chronic illnesses or conditions requiring ongoing medical treatment. These women self-report being in generally good health, with only minor or insignificant health concerns that do not interfere with their daily activities.
9.
Response: Thank you for your review. We noticed that no specific comment was provided for this point. If there are any concerns or suggestions regarding this section, we would be happy to address them.
- Table 1
- a) decimal point not comma for the numbers eg 0,001 should be 0.001. All
numbers need to be corrected.
- b) You have in the legend under the table n=p= sample size.
Change this to n as the column label and in the legend n=number of
participants
- c) check VO2 in measurement column – for BJG you have V’02 – needs to be
corrected
Response: Thank you for your suggestions. Corrections have been made as follows:
- a) The decimal point has been used instead of a comma for all numbers (e.g., 0.001 instead of 0,001). All numbers have been corrected accordingly.
- b) The legend under the table has been updated. The phrase "n=p= sample size" has been corrected to "n" as the column label, and in the legend, "n= number of participants" is now correctly stated.
- c) The VO2 notation in the measurement column for BJG has been corrected to the proper format.
11.Line 242 – move % to be after 0.57 not after group
Response: Thank you for your suggestion. The correction has been made as follows: The results showed that BJG VO₂ max increased by 4.82% (from 35.24 ± 5.07 to 36.94 ± 4.91 ml·min⁻¹·kg⁻¹), while it decreased by 0.57% (from 35.06 ± 4.87 to 34.86 ± 5.01 ml·min⁻¹·kg⁻¹) in the PLG group between the first and second tests (see Figure 3).
- Line 245 – remove “of”. Just have - after consuming 50 ml etc
Response: Thank you for your suggestion. The correction has been made as follows: "of" has been removed from line 245. The sentence now reads: "after consuming 50 ml..."
13.Was there a significant increase seen in BJ V02 max? In table 1 shows trending
towards significance. For Placebo there is a statistically significant change
Figure 3 should be removed as it is not showing anything and you can have in the
text the information about the significance of the changes.
Response: Thank you for your suggestions. Corrections have been made as follows:
- There was a trend towards a significant increase in BJ VO₂ max. Table 1 illustrates this trend.
- A statistically significant change was observed for the placebo group.
- Figure 3 has been removed as it did not provide meaningful data.
- Lines 247-249 have been rewritten for clarity.
14.Line 247-249 – rewrite and tidy up. E.g
Statistically significant changes in mean heart rate in BJP and PLG after test 1
and test 2 were observed and are shown in Figure 4. Mean heart rate (HR) in the
BJG group changed significantly after the second test (p < 0.05), decreasing from
165 247 ± 9 to 162 ± 10 bpm. On the other hand, in the PLG group, HR increased
from 160 ± 9 to 164 ± 8 beats per minute after the first and second tests.
Response: Thank you for your suggestions. Corrections have been made as follows:
- Lines 247-249 have been rewritten.
- Statistically significant changes in mean heart rate in BJG and PLG after Test 1 and Test 2 were observed and are shown in Figure 4.
- Mean heart rate (HR) in the BJG group changed significantly after the second test (p < 0.05), decreasing from 165 ± 9 to 162 ± 10 bpm.
- In contrast, in the PLG group, HR increased from 160 ± 9 to 164 ± 8 bpm after the first and second tests.
15.Line 252 – remove “of” – just have after consuming 50 ml etc
Response: Thank you for your suggestion. The correction has been made as follows: "of" has been removed from line 252
16.Lines 254 – 257 – rewrite. Same suggestion as in point 12 above.
Response: Thank you for your suggestions. Corrections have been made as follows:
- Lines 254-257 have been rewritten, as per your suggestion in point 12 above.
- The phrase "of" has been removed from lines 257-264.
17.Line 259 - remove “of”. Just have - after consuming 50 ml etc [ make sure other
legends for other figures are corrected]
Response: Thank you for your suggestions. The corrections have been made as follows:
- "Of" has been removed from line 259.
- The legends for other figures have also been corrected.
18.Line 269 – you have the cohen value d=0.9 in brackets. A sentence on the effect
size of this change is important and should be stated .. is it small effect, medium
effect? What about the other aerobic capacity parameters you have measured?
What are the effect size changes with these?
Response: Thank you for your suggestions. Corrections according to them were made: In the BJG group, mean VE/VCO₂ values decreased significantly (p = 0.025) from 0.030 ± 0.00 to 0.028 ± 0.00 (d = 0.9) The effect size of this change is important it is large (0.8) effect (Cohen, 2013).
Discussion
Line 287 – there was an increase but it is not statistically significant – it is
trending towards significance?
Line 287- you say “indicating to improved oxygen etc”-- What do you mean to
improved?
The section and paragraphs discussing VO2 max needs to be altered slightly to
reflect the results of your study.
Line 293 – remove comma after eg and have full stop. Check in rest of document for these type of typos. Line 301 - Full stop after [4,5] Line 303 – strong words eg “significant factor” as the results in this study do not show that Line 320 321 – “On the contrary” – in this context probably better to say “in contrast” Line 335 – don’t need arguably?
Response: Thank you for your valuable feedback. We appreciate your detailed suggestions and have made the necessary revisions to address the points you raised.
- Line 287 - Increase not statistically significant: We agree with your comment. There was a trend towards significance in the data, but it did not reach statistical significance. The text has been revised to reflect this: "There was a trend towards significant changes in maximal oxygen consumption (VO₂ max), which increased by 4.82% in the BJG group (from 35.24 ± 5.07 to 36.94 ± 4.91 ml·min⁻¹·kg⁻¹)."
- Line 287 - "Indicating improved oxygen availability": The phrase has been clarified. We now state that the increase in VO₂ max suggests "improved oxygen utilization and enhanced blood flow to muscles."
- Line 293 – Comma after "e.g.": The comma after "e.g." has been removed and replaced with a full stop, as requested.
- Line 301 – Full stop after [4,5]: A full stop has been added after the citation [4,5].
- Line 303 – Strong words like "significant factor": We have reworded this part to reflect the actual findings of the study. The phrase now reads: "This study suggests that [factor] may influence..." instead of calling it a "significant factor."
- Line 320-321 – "On the contrary" vs. "In contrast": We have revised the text to use "In contrast" for better clarity and appropriateness in this context.
- Line 335 – "Arguably": We have removed "arguably" from this section to strengthen the statement and improve precision.
We believe these changes improve the clarity and accuracy of the manuscript, and we appreciate your help in improving the overall quality of the work.
Other questions:
- a) How do your results compare to studies that have male athletes? A big factors in regard to your study was emphasising there could be difference sin females due to hormonal differences. Were there any when compared to similar studies with only male participants? b) Line 387- Doesn’t your study support other studies that have also shown BJ can enhance performance during intense physical activity?
Response: a) Thank you for your question. Our study focused on female athletes, and while hormonal differences may affect exercise performance, the findings from studies on male athletes can provide valuable insight. However, based on the existing literature, it is difficult to directly compare results across genders, as there are fewer studies examining the effects of beetroot juice (BJ) supplementation in male athletes specifically. Most research on BJ and exercise performance has involved mixed-gender populations or male participants, and results in male athletes may vary depending on training status, exercise modality, and BJ dosage. We encourage further research focusing on male athletes to determine if hormonal differences contribute to gender-specific outcomes.
- b) Thank you for your comment. Yes, our study does support the existing body of research that has demonstrated the potential benefits of BJ supplementation for enhancing performance during intense physical activity. Our results align with studies showing improvements in parameters like VO₂ max, heart rate, and ventilatory efficiency, which are consistent with BJ’s role in improving exercise capacity and endurance. While our study adds valuable insights into how BJ may specifically affect female athletes, it also supports the broader understanding that BJ supplementation can enhance performance in various populations during high-intensity exercise.
We would like to sincerely thank you for your valuable feedback and insightful comments, which have greatly contributed to the improvement of our manuscript. Your suggestions have been extremely helpful in enhancing the clarity and quality of the paper.
As a result of the changes made in response to your comments, there may be slight discrepancies in the line numbers. To facilitate the review of the revised manuscript, all corrections have been highlighted in the attached document.
Once again, we appreciate your time and effort in reviewing our work and providing thoughtful suggestions.
Baranauskas, M. N., Freemas, J. A., Tan, R., & Carter, S. J. (2022). Moving beyond inclusion: Methodological considerations for the menstrual cycle and menopause in research evaluating effects of dietary nitrate on vascular function. Nitric Oxide, 118, 39-48. doi:10.1016/j.niox.2021.11.001
Cohen, Jacob. (2013). Statistical Power Analysis for the Behavioral Sciences (Rev. ed.): Academic Press.
Hamilton, L. J. (2006). Comment on: “Orpin, A.R. and Kostylev, V.E., 2006. Towards a statistically valid method of textural sea floor characterization of benthic habitats [Mar. Geol. 225 (1–4), 209–222.]”. Marine Geology, 232(1), 105-110. doi:https://doi.org/10.1016/j.margeo.2006.08.001

Reviewer 2 Report
Comments and Suggestions for Authors
At the research methodology please describe the subjects activities other than sport activities and also the schedule of the sport activities during day/week and month.
Please at the discussion section display how the cited references are situated in comparison with your study ... Present range of values and findings and also comment how this are related or in contradiction with your results.
Author Response
For research article
Response to Reviewer Comments
|
||
1. Summary |
|
|
Thank you very much for taking the time to review this manuscript. Please find the detailed responses below and the corresponding revisions/corrections highlighted in the re-submitted files.
|
||
2. Questions for General Evaluation |
Reviewer’s Evaluation |
Response and Revisions |
Does the introduction provide sufficient background and include all relevant references? |
Yes |
|
Are all the cited references relevant to the research? |
Yes |
|
Is the research design appropriate? |
Yes |
|
Are the methods adequately described? |
Yes |
|
Are the results clearly presented? |
Can be improved |
|
Are the conclusions supported by the results? |
Can be improved |
|
Comments and Suggestions for Authors
Comments: At the research methodology please describe the subjects activities other than sport activities and also the schedule of the sport activities during day/week and month.
Response: Thank you for your valuable feedback. In response to your comment, I have updated the methodology to include a more detailed description of the participants' activities outside of structured sports, as well as their weekly and monthly sport activity schedule. The revised section now outlines their sedentary lifestyle outside of training, with a focus on their academic responsibilities. Additionally, the specific training volume, duration, and intensity have been clarified. I hope this added information addresses your concern.
‘’The participants were mainly university students with a sedentary lifestyle outside structured sports. They did not engage in other recreational or physically demanding activities, ensuring consistent physical activity levels. Their daily schedules were focused on academic responsibilities. They practiced endurance sports during their free time, with weekly training volumes ranging from 150 to 300 minutes in the moderate-intensity heart rate zone. Warm-up and cool-down periods made up about 30% of the total session time but were not included in the target zone. The participants had adopted this training regimen 3 months to 1 year before the study, ensuring they were still in the adaptation phase of endurance training, minimizing the impact of long-term, high-intensity training on results.’’
Comments: Please at the discussion section display how the cited references are situated in comparison with your study ... Present range of values and findings and also comment how this are related or in contradiction with your results.
Response: Thank you for your valuable feedback and for highlighting the need to contextualize our findings within the existing literature. We have revised the discussion section to explicitly display how the cited references compare to our study, including presenting a range of values and findings and commenting on their relevance or contradictions with our results.
Specifically, we have:
- Compared VO₂ max results:
- We noted alignment with the findings of [1], which documented improved VO₂ max and oxygen economy in response to dietary nitrates. This study included both men and women, whereas our study focused on female endurance athletes, adding a gender-specific perspective.
- We referenced [2], which similarly documented VO₂ max improvements in participants with moderate aerobic capacity, supporting the generalizability of our results.
- Additionally, we acknowledged contrasting findings from [3], which observed less pronounced VO₂ max improvements among elite athletes, potentially due to physiological saturation effects. This contextualizes our findings within the spectrum of athletic performance levels and highlights the relative benefits for individuals with moderate fitness levels, as observed in our BJG group.
- Explored reductions in VE:
- We related our findings to [4], which emphasized dietary nitrates' role in enhancing oxygen utilization and reducing respiratory strain during high-intensity exercise. This comparison underscores the broader consistency in VE-related benefits of nitrate supplementation across different exercise modalities and populations.
These revisions provide a clearer situational context for the cited references, demonstrating how they align with or differ from our study outcomes. We hope these changes adequately address your concerns and enhance the clarity and depth of our discussion.
Thank you again for your insightful comments.
- Bailey, S.J., et al., Dietary nitrate supplementation reduces the O2 cost of low-intensity exercise and enhances tolerance to high-intensity exercise in humans. J Appl Physiol (1985), 2009. 107(4): p. 1144-55.
- Lansley, K.E., et al., Acute dietary nitrate supplementation improves cycling time trial performance. Med Sci Sports Exerc, 2011. 43(6): p. 1125-31.
- Hoon, M.W., et al., Nitrate supplementation and high-intensity performance in competitive cyclists. Appl Physiol Nutr Metab, 2014. 39(9): p. 1043-9.
- Jones, A.M., et al., Dietary Nitrate and Physical Performance. Annu Rev Nutr, 2018. 38: p. 303-328.
